Research

bioinformatics/plant science

AtHsp90, plant embryogenesis, conformational dynamics

# Identification of AtHsp90.6 involved in early embryogenesis and its structure prediction by molecular dynamics simulations

An Luo[1], Xinbo Li[2,3], Xuecheng Zhang[2], Huadong Zhan[4], Hewei Du[1], Yubo Zhang[5] and Xiongbo Peng[2]

[1]College of Life Science, Yangtze University, Jingzhou 434023, People's Republic of China
[2]College of Life Science, State Key Laboratory of Hybrid Rice, Wuhan University, Wuhan 430072, People's Republic of China
[3]Center for Tissue Engineering and Regenerative Medicine, Union Hospital, Tongji Medical College, Huazhong University of Science and Technology, Wuhan 430072, People's Republic of China
[4]College of Life and Environment Sciences, Shanghai Normal University, Shanghai, 200234, People's Republic of China
[5]Department of Food Science, Foshan University, Foshan 528231, People's Republic of China

 AL, 0000-0001-8405-5241; YZ, 0000-0002-7762-0027;
XP, 0000-0002-9436-5072

**Authors for correspondence:**
Yubo Zhang
e-mail: zhyubo7@gmail.com
Xiongbo Peng
e-mail: bobopx@whu.edu.cn

Heat-shock protein of 90 kDa (Hsp90) is a key molecular chaperone involved in folding the synthesized protein and controlling protein quality. Conformational dynamics coupled to ATPase activity in N-terminal domain is essential for Hsp90's function. However, the relevant process is still largely unknown in plant Hsp90s, especially those required for plant embryogenesis which is inextricably tied up with human survival. Here, AtHsp90.6, a member of Hsp90 family in *Arabidopsis*, was firstly identified as a protein essential for embryogenesis. Thus we modelled AtHsp90.6 in its functionally closed 'lid-down' and open 'lid-up' states, exploring the nucleotide binding mechanism in these two states. Free energy landscape and electrostatic potential analysis revealed the switching mechanism between these two states. Collectively, this study quantitatively analysed the conformational changes of AtHsp90.6 bound to ATP or ADP. This result may help us understand the mechanism of action of AtHsp90.6 in future.

# 1. Introduction

Hsp90 (heat-shock protein 90) is a kind of ATP-dependent molecular chaperone required for correct maturation and activation of client proteins. It is involved in diverse cellular processes, for instance, signal transduction, protein trafficking, protein degradation and maintaining protein homeostasis [1]. Deletion of Hsp90 would impact stress responses, growing development, and of course embryogenesis [2] which is significant for development of animals and plants. Especially for plants, embryo not only forms a basic pattern of plant body, but also is an important component during the formation of seed [3], which provides an irreplaceable material base for human survival.

Structurally, Hsp90 contains three distinct domains: the N-terminal ATPase domain, the middle domain responsible for client proteins binding and contains catalytically vital residue needed for ATP hydrolysis, and the C-terminal dimerization domain [4]. Without nucleotide, constitutive dimerization of HSP90 was formed by the C-terminal domain, reaching an 'open conformation'. While, ATP binding made Hsp90 be dimerized at both the N- and C-terminal domains to form a 'close conformation' that is efficient for ATP hydrolysis [4]. In the conformational and ATPase cycle, ATP, co-chaperone and client protein were recruited by Hsp90 to form functional chaperone, and then ADP, co-chaperone and properly folded client protein were released with Hsp90 returning to the 'open conformation' [4]. In addition, a 'lid' segment formed by certain conserved amino acid residues in the N-terminal ATPase domain opened during the ADP-bound state and closed over the nucleotide binding pocket in the ATP-bound state [5,6]. Thus ATP binding and hydrolysis in N-terminal ATPase domain directly concerns the biological activity and function of Hsp90 [2].

Although the mechanism by which Hsp90s bind and hydrolyse ATP is already researched in yeast and mammals [4,7], the fact that Hsp90s exhibited different nucleotide-binding affinities among different species such as dog and yeast [8] suggested the necessity for figuring out the individual feature of different Hsp90s involved in diverse functions or species. Recently, several studies also focused on the structure of the N-terminal domain in plant Hsp90s [1,9]. However, the knowledge of conformational changes associated with ATP/ADP binding in N-terminal domain of plant Hsp90s is still limited, let alone those of Hsp90s required for plant embryogenesis. Therefore, it instantly raises a question: what is the molecular level detail of the process in the Hsp90s required for plant embryogenesis?

As the genome of *Arabidopsis*, a model plant for the study of embryogenesis [10] has already been sequenced, seven members of *AtHsp90* family were known to be located in different chromosomes. By genetic manipulation, we firstly identified that *AtHsp90.6* was essential for embryogenesis. Then we explained how AtHsp90.6 recognized the nucleotide by its N-terminal region. Our structural analysis suggested AtHsp90.6 had its functionally closed 'lid-down' and open 'lid-up' states. We then explored the nucleotide binding mechanism in these two states. Additionally, we investigated the switching mechanism by free energy landscapes and electrostatic surface potentials. This result may help us understand how the ATPase cycle of AtHsp90.6 is coupled to client protein activation in future.

# 2. Material and methods

## 2.1. Plant materials and growth conditions

The *Arabidopsis thaliana* ecotype Columbia (Col-0) was used as the wild-type. The *athsp90.6-1* mutant was screened out from our mutant library [11]. The *athsp90.6-2* mutant (SALK_021119) was obtained from the ABRC (Arabidopsis Biological Resource Center, https://abrc.osu.edu/). After surface sterilized, seeds were sown on 1/2 MS plates with 1.0% (w/v) sucrose and proper antibiotics as previous methods described [11]. Seedlings were then transplanted to vermiculite in a greenhouse. The growth conditions were at 22°C under a 16 h light period.

## 2.2. Characterization of the T-DNA flanking sequence of the *athsp90.6-1* and *athsp90.6-2*

Thermal asymmetric interlaced PCR (TAIL-PCR) was used to clarify the T-DNA flanking sequence of the mutants [12]. Primers located around the T-DNA border were designed to determine the T-DNA insertion site. *athsp90.6-1*: (RP1:TTCGTTTTGAGCAAATGGTTAGTAG, LP1:CCTTCTTGGTCTTCTTCT TTTTCTG, LB-S: CCAAAATCCAGTACTAAAATCCAG). *athsp90.6-2*: (LP2: TCTCTGCATCGTGAGAA TGTG, RP2: TGCTGGAGAAAGGACTTGAAG, BP: ATTTTGCCGATTTCGGAAC).

## 2.3. Vector construction and plant transformation

Plasmid P092 and P095 were obtained in previous work [11]. To generate the *proAtHsp90.6::AtHsp90.6* complementation construct, a 6255 bp wild-type genomic sequence containing the *AT3g07770* gene was cloned into the P092. Relevant primers were *AtHsp90.6*-F1: NNNNGGTACCTGCACTAGGAAG AAACAAACTCAAG and *AtHsp90.6*-R1: NNNNCCTAGGATTCGTACGATAACAACATCCCTC. To detect the expression pattern of *AtHsp90.6*, its promoter was amplified and cloned into the P095 to generate the *proAtHsp90.6::H2B-GFP* (green fluorescent protein) construct. Relevant primers were *AtHsp90.6*-F1 and *AtHsp90.6*-R2: NNNNGAATTCCGTCGCAAACTTCTAAAATCTCG.

The two constructs mentioned above were mobilized into *Agrobacterium tumefaciens* strain GV3101 by electrotransformation, respectively. Then the floral dip method was used for transgene in *Arabidopsis* plants [13].

## 2.4. Phenotype characterization of embryogenesis

To compare the embryogenesis among the *athsp90.6-1*, *athsp90.6-2* mutants and the wild-type plant, the whole-mount clearing technology was applied as previously described [11]. Images were processed with Adobe Photoshop.

## 2.5. Ovule separation and confocal laser scanning microscope (CLSM) microscopy

Ovules at specific development stages were manually dissected from the siliques by using a sharp capillary glass tube. Isolated ovules were collected and put in a 30 mm diameter culture plate with a drop of 10% mannitol. GFP fluorescence of ovules was then detected through a FV1000 confocal laser-scanning microscope. Images were processed with Adobe Photoshop.

## 2.6. Generation of atomic models of AtHsp90.6

The modelling process was performed as our previous descriptions [14,15]. The amino acid sequence of AtHsp90.6 from *Arabidopsis thaliana* was obtained from the NCBI (Gene ID: 819968). In the first homology modelling step, template structures related to the AtHsp90.6 protein were searched against the whole Protein Data Bank (PDB) using the Blast algorithm [16]. Given the sequence identity between our model and the crystal structure (PDB ID: 2CG9) [6] was 56%, we homolgly modelled AtHsp90.6 by our established methods [15,17,18]. SWISS-MODEL is a web-based integrated service dedicated to protein structure homology [19], and we used the default parameters to generate the model.

## 2.7. All-atom molecular dynamics simulations

The 100 ns all-atom molecular dynamics (MD) simulations were performed with the GROMACS 4.5.3 software package [20] using the ff99 force field [21] and the TIP3P [22] water model as in our previous reports [23]. The protonation state of ionizable groups was chosen to correspond to pH 7.0. Counterions were added to compensate the net charge of the system.

The parameters for ADP and ATP were taken from the AMBER parameter database, maintained by The Bryce Group (http://research.bmh.manchester.ac.uk/bryce/amber). The parameters were developed by Carlson *et al.* [24], and their details have been put in the electronic supplementary material. To perform MD simulations with the GROMACS software package, we conduct the conversion to GROMACS compatible topology using ACPYPE [25].

The initial structure of N-terminal AtHsp90.6 was immersed in a periodic water box. The electrostatic interactions were calculated by using the particle-mesh Ewald (PME) algorithm [26], and the van der Waals forces were treated with a cut-off distance of 10 Å. After 3000 steps of energy minimization using steepest descent method, the system was subject to 300 ps of equilibration at 300 K and normal pressure, using harmonic position restraints with a force constant of $1000 \, \text{kJ mol}^{-1} \, \text{nm}^{-2}$. The system was coupled to an external bath by the Berendsen pressure and temperature coupling method [27]. The production run was performed under the same conditions except that all position restraints were removed. The results were analysed using the standard software tools provided by the GROMACS package. The in-house analysis scripts and protocols were performed as in our previous literature [15,17,18,28–30]. Visualization and manipulation of the conformations was performed using the Visual Molecular Dynamics program [31]. The statistical analysis was performed using the R statistical software package [32].

## 2.8. Coarse-grained MD simulations

The 1000 ns coarse-grained (CG) MD simulations were performed as described by Marrink *et al.* [33] and our group [14]. The atomistic model of full length AtHsp90.6 was converted to CG model. The MARTINI forcefield [34] was used for the CG-MD simulations. Eighteen CG particle types were divided into four categories: polar (P), charged (Q), mixed polar/apolar (N) and hydrophobic apolar (C). The Lennard-Jones interactions were used for the inter-particle interactions, including 10 subtypes to reflect the hydrogen bonding capabilities or polarity. The elastic network modelling was built across the CG backbone beads, aiming to maintain the protein secondary structure. The parameters of ATP were taken from [35] with minor modifications. Each ATP was mapped to nine CG beads. Three beads represented phosphate ester and six beads represented sugar and base, respectively. Note that the adenine purine was modelled with four-bead rings.

All CG-MD simulations were performed using GROMACS 4.5.3 [20]. The Berendsen weak coupling algorithm was used to maintain the temperature (coupling constant of 0.5 ps; reference temperature 300 K) and pressure (coupling constant of 1.2 ps; reference pressure 1 bar), respectively. The non-bonded interactions were treated with a switch function from 0.0 to 1.2 nm for the Coulomb interactions, and from 0.9 to 1.2 nm for the Lennard-Jones interactions. The integration time-step was set to 20 fs and the neighbour list was updated every five steps. The system was solvated in a rectangular box with a minimum of 1.0 nm between any protein bead and the edge of the box. After energy-minimization with position restraints ($1000\ \mathrm{kJ\ mol^{-1}\ nm^{-2}}$) applied to all protein beads, a 50 ps MD simulation using a 1 fs time-step with the same position restraints was used to relax both the solvent molecules and the protein. The system was further relaxed with 1 ns long MD run, using a 20 fs time-step and position restraints on the protein 'backbone' beads. Finally the system was simulated 1000 ns for production without any restraints. The cut-off value of the number of contacts were 0.7 nm according to Sansom's method [36,37].

## 2.9. MD simulations and calculation of binding free energies by MM-GBSA

We carried out MD simulations to further investigate the protein–ligand interaction for ADP-AtHsp90.6, ATP-AtHsp90.6, the experimental structures PDB IDs: 5FWK [38] and 4XCJ [39]. The binding free energies were then recalculated using the MD trajectories.

To perform MD simulations, 1000 ps GB–MD (IGB = 2) unrestrained simulations were performed on all the complexes with a time-step of 2.0 fs. The generalized Born (GB) solvation model in macromolecular simulations [40] was used instead of explicit water. The parameters for ADP and ATP were developed by Carlson *et al.* [24]. Temperature was set at 300 K with the cut-off distance of 12 Å used for non-bonded interaction. Before the unrestrained MD simulations was performed, we employed enough equilibration steps for 500 ps from a larger force constant $5.0\ \mathrm{kcal\ mol^{-1}\ \mathring{A}^{-2}}$ for restraining all heavy atoms and then gradually reduced it to $0.02\ \mathrm{kcal\ mol^{-1}\ \mathring{A}^{-2}}$ for only heavy atoms in the backbone. Throughout all the energy minimization and simulation processes, the ff99 force field (Parm 99) [21] was applied.

The binding free energy was estimated by the MM-GBSA approach [41–43]. In the MM-GBSA implementation of Amber 11.0 [44], the binding free energy of $A + B \rightarrow AB$ is calculated using the following thermodynamic cycle:

$$
\begin{array}{ccccc}
A_{\mathrm{aqu}} & + & B_{\mathrm{aqu}} & \xrightarrow{\Delta G_{\mathrm{binding}}} & AB_{\mathrm{aqu}} \\
\Big\downarrow {\scriptstyle -\Delta G_{\mathrm{solv}}^{A}} & & \Big\downarrow {\scriptstyle -\Delta G_{\mathrm{solv}}^{B}} & & \Big\downarrow {\scriptstyle -\Delta G_{\mathrm{solv}}^{AB}} \\
A_{\mathrm{gas}} & + & B_{\mathrm{gas}} & \xrightarrow{\Delta G_{\mathrm{gas}}} & AB_{\mathrm{gas}}
\end{array}
$$

$$
\begin{aligned}
\Delta G_{\mathrm{binding}} &= \Delta G_{\mathrm{gas}} - \Delta G_{\mathrm{solv}}^{A} - \Delta G_{\mathrm{solv}}^{B} + \Delta G_{\mathrm{solv}}^{AB} \\
&= \Delta H_{\mathrm{gas}} - T\Delta S - \Delta G_{\mathrm{GBSA}}^{A} - \Delta G_{\mathrm{GBSA}}^{B} + \Delta G_{\mathrm{GBSA}}^{AB} \\
&= \Delta H_{\mathrm{gas}} - T\Delta S + \Delta\Delta G_{\mathrm{GB}} + \Delta\Delta G_{\mathrm{SA}}
\end{aligned}
$$

$$
\Delta H_{\mathrm{gas}} \approx \Delta E_{\mathrm{gas}} = \Delta E_{\mathrm{intra}} + \Delta E_{\mathrm{elec}} + \Delta E_{\mathrm{vdw}}
$$

$$
\Delta\Delta G_{\mathrm{GB}} = \Delta G_{\mathrm{GB}}^{AB} - (\Delta G_{\mathrm{GB}}^{A} + \Delta G_{\mathrm{GB}}^{B})
$$

$$
\Delta\Delta G_{\mathrm{SA}} = \Delta G_{\mathrm{SA}}^{AB} - (\Delta G_{\mathrm{SA}}^{A} + \Delta G_{\mathrm{SA}}^{B})
$$

where $T$ is the temperature, $S$ is the solute entropy, $\Delta G_{\text{gas}}$ is the interaction energy between A and B in the gas phase, and $\Delta G_{\text{solv}}^{A}$, $\Delta G_{\text{solv}}^{B}$ and $\Delta G_{\text{solv}}^{AB}$ are the solvation free energies of A, B and AB, which are estimated using a GB surface area (GBSA) method.

That is, $\Delta G_{\text{solv}}^{AB} = \Delta G_{\text{GBSA}}^{AB} + G_{\text{GB}}^{AB} + \Delta G_{\text{SA}}^{AB}$, and so forth. $\Delta G_{\text{GB}}$ and $\Delta G_{\text{SA}}$ are the electrostatic and non-polar terms, respectively. The bond, angle and torsion energies constitute the intramolecular energy $\Delta E_{\text{intra}}$ of the complex, while $\Delta E_{\text{elec}}$ and $\Delta E_{\text{vdw}}$ represent the receptor–ligand electrostatic and van der Waals interactions, respectively. We refer to $\Delta G_{\text{binding}}^{*}$ for $\Delta G_{\text{binding}} + T\Delta S$ in the discussion as suggested by previous studies [17,23]. Given previous MM-GBSA studies [45–47] showed 30, 40 and 100 snapshots and short simulation time could give reasonable results in correlation with experimental data, we collected 50 snapshots at the final equilibrium stage for calculation.

## 2.10. Free energy landscape

The energy landscape of the dynamics behaviour of AtHsp90.6 was estimated by an appropriate sampling method [23]. In order to get a two-dimensional energy landscape map, RMSD (root mean square deviation) and Rg (radius of gyrate) of AtHsp90.6 were chosen as two reaction coordinates. The energy landscape was calculated along these two reaction coordinates using the equation [17],

$$\Delta G(p1, p2) = -\kappa_{B} T \ln \rho(p1, p2),$$

where $\kappa_{B}$ represent the Boltzmann constant, $T$ is the simulated temperature, and $\rho(p1, p2)$ represent the normalized joint probability distribution.

## 2.11. Electrostatic potential calculations

Electrostatic potential maps were calculated with the adaptive Poisson–Boltzmann solver (APBS) [48] according to default parameters (physiological salt concentration of 150 mM, temperature of 298.15 K, solvent dielectric of 78.54 and solute dielectric of 2). Solute charges were distributed onto grid points using a cubic B-spline discretization. The molecular surface was defined by the interface between the radius of a water molecule (1.4 Å), and the solute van der Waals radii.

## 2.12. Statistical analyses

All data are presented as the mean $\pm$ s.e. SPSS19 software [49] was used for statistical analyses. Significant difference was estimated by an independent $t$-test or $\chi^2$ test. Tests are two-tailed. $p < 0.05$ was considered significant.

# 3. Results

## 3.1. Phenotype identification of *athsp90.6* by mutant analysis experiment

*AtHsp90.6*, a single-copy gene, was certified to locate on *Arabidopsis* chromosome III (accession number: AT3g07770) (figure 1*a*). By using our established mutant library [11], we screened out a T-DNA insertion *athsp90.6* mutant, named as *athsp90.6-1*, from approximately 7600 mutants. An allelic T-DNA insertion mutant named as *athsp90.6-2* was obtained from ABRC respectively (figure 1*a*).

The siliques both from the $F_1$ generation of *athsp90.6-1* and *athsp90.6-2* heterozygous mutants, contained green ovules and albino ovules, while the siliques from the wild-type only contained green ovules (figure 1*b–d*). The seed set ratios of the two heterozygous mutants were about 74% and 72% ($n = 877$, 906), obviously lower than that of the wild-type (figure 1*f*). Given the fact that neither *athsp90.6-1* nor *athsp90.6-2* homozygous mutants could be obtained (electronic supplementary material, figure S1), it suggested that loss function of *AtHsp90.6* may lead to embryo lethality. Moreover, it showed that the pattern formation of embryos in albino ovules was normal during early embryogenesis, but the growth of the embryos arrested in the latter stages. When embryos in green ovules had been at the cotyledon stage, embryos in albino ovules from the same silique remained at the globular-embryo stage (figure 1*g–p*).

To further confirm the defects in the two mutants were indeed caused by the loss function of *AtHsp90.6*, a genomic *AtHsp90.6* of the wild-type was transformed to the *athsp90.6-1* heterozygous mutant. Indeed, homozygous *athsp90.6-1* mutant could be obtained in the $T_2$ generation of

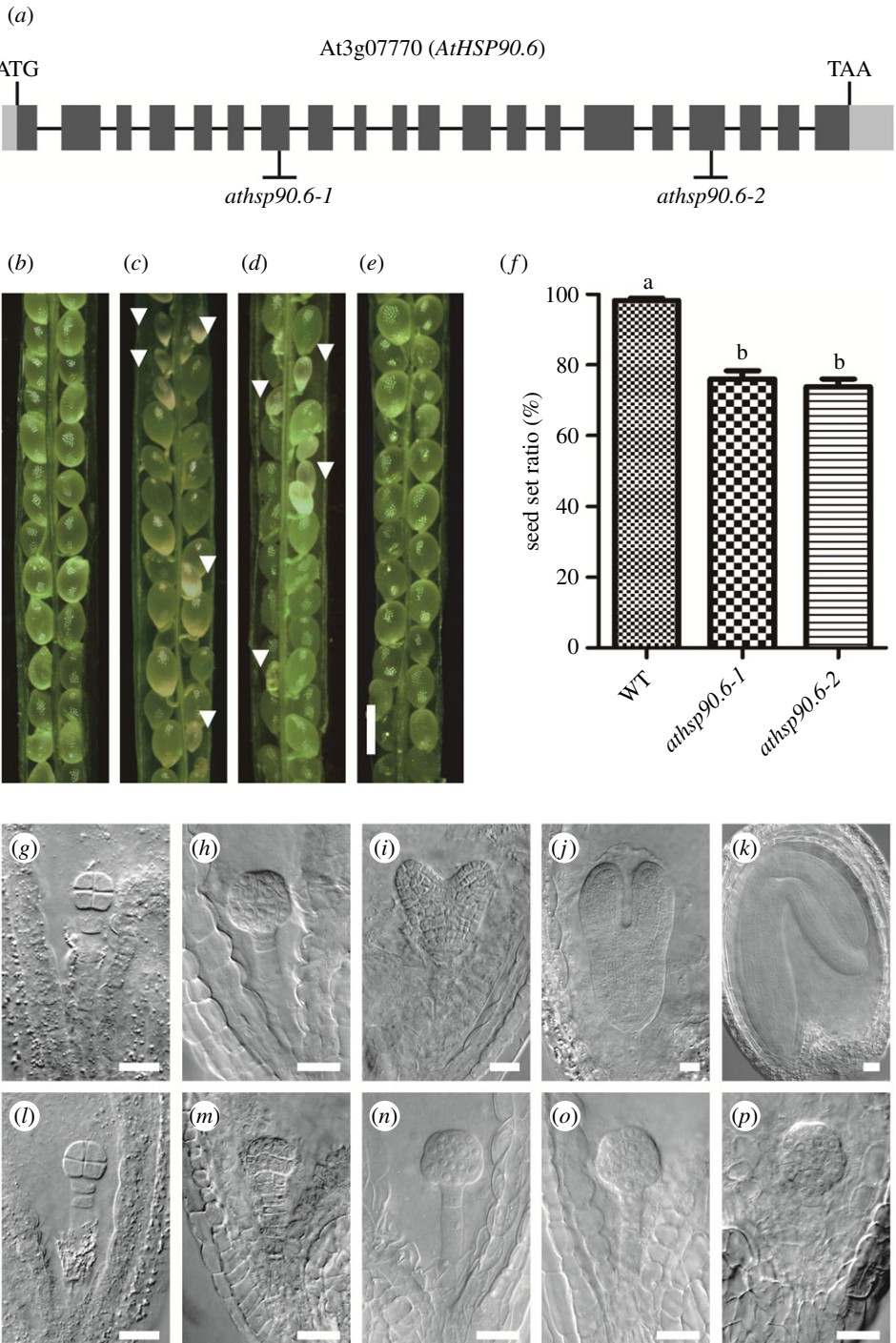

**Figure 1.** Embryogenesis is defective in *athsp90.6* plant. (*a*) Diagram of the *AtHsp90.6* genomic DNA and the T-DNA insertion sites. The *AtHsp90.6* gene (AT3g07770) has 20 exons. The T-DNA inserted into the seventh exon in *athsp90.6-1* and the 17th exon in *athsp90.6-2*. (*b*) Image of silique of wild-type plant. (*c*) Image of silique of *athsp90.6-1* plant. (*d*) Image of silique of *athsp90.6-2* plant. (*e*) Image of silique of the complementation line. White aborted seeds indicated by arrow head, bar = 500 μm. (*f*) Seed set ratio of wild-type (WT) plant and *athsp90.6-1* and *athsp90.6-2* mutant. Data are expressed as the mean ± s.e., $n = 840\ 877\ 906$. Columns not sharing the same small letter means significant different ($p < 0.05$). (*g–k*) Normal development of embryo in green ovules (from the 8-celled embryo stage to the cotyledon stage). Bar = 20 μm. (*l–p*) Arrested development of embryo in albino ovules. Bar = 20 μm. Embryos in *g* and *l*, *h* and *m*, *i* and *n*, *j* and *o*, *k* and *p* were from the same silique in *athsp90.6-1* plant, respectively.

complementation lines (*athsp90.6-1*/*athsp90.6-1*, g-AT3g07770/g-AT3g07770). In addition, seed development began to normalize in the complementation lines (figure 1*e*; electronic supplementary material, table S1). Collectively, these data revealed that *AtHsp90.6* plays a significant role during embryogenesis. All methods and procedures we adopted above are proved to be reliable [11].

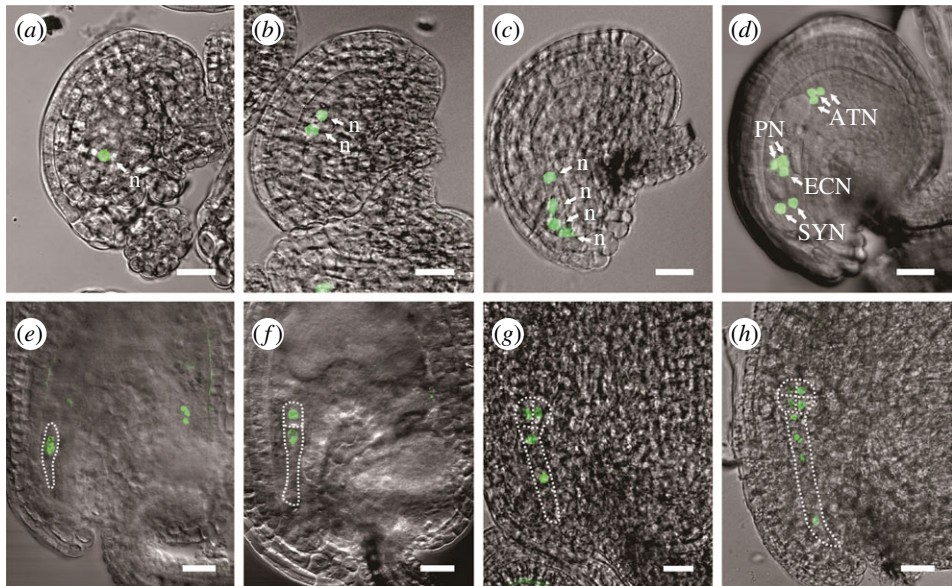

**Figure 2.** *AtHsp90.6* expression analysis during female gametogenesis and early embryogenesis. (*a–d*) *proAtHsp90.6::H2B-GFP* expressed in the different stage of female gametogenesis: 1-nucleate embryo sac (*a*); 2-nucleate embryo sac (*b*); 4-nucleate embryo sac (*c*); 8-nucleate embryo sac (*d*). (*e–h*) *proAtHsp90.6::H2B-GFP* expressed in early embryogenesis: zygote (*e*); two-celled proembryo with an apical and a basal cell (*f*); proembryo with divided apical cell (*g*); 8-celled embryo (*h*). The dashed lines indicate the edges of the embryos. Bar = 20 μm. The arrows indicate the nuclei with GFP signal. ECN, egg cell nucleus; SYN, synergid cell nuclei; ATN, antipodal cell nuclei; PN, polar nuclei; n, nucleus.

## 3.2. Expression pattern analysis of *AtHsp90.6* by fluorescence observation experiment

A 1.5 kb fragment of the *AtHsp90.6* native promoter was fused to reporters H2B-GFP to produce the *proAtHsp90.6::H2B-GFP* transgenic lines [50]. The transgenic $T_2$ progeny of *proAtHsp90.6::H2B-GFP* homozygous lines showed that the GFP fluorescence could be only detected in female gametophyte and early embryo, but not in other tissues of the ovule (figure 2). The specific expression pattern of *AtHsp90.6* in the ovule is strongly consistent with the phenotype caused by loss function of *AtHsp90.6*. Since such an important protein is found, we are interested in its structural information for further understanding this protein.

## 3.3. *In silico* structural analysis of AtHsp90.6

Due to the missing structural information for AtHsp90.6, we homologically modelled the dimer of AtHsp90.6. The structure of AtHsp90.6 can be divided into three regions: the N-terminal (M1-P385), middle (L386-D613) and the C-terminal (S614-K799) regions (figure 3*a*). Observably, the N-terminal domain had an ATP-binding region, containing a lid represented by residues G185 to G216. The lid was composed of three loops L1, L2 and L3.

This whole structure was named as AtHsp90.6 FL and is shown in electronic supplementary material, table S2. Given the nucleotide can directly bind to the N-terminal of AtHsp90.6, we select the N-terminal for the nucleotide binding study and conformation dynamics study. The N-terminal of AtHsp90.6 was named as AtHsp90.6N.

To assess the generated model, we adopted two criteria. Firstly, the structural similarity between our model and the crystal structure was measured by the root mean square deviation (RMSD) in their best-superimposed atomic coordinates. The RMSD for AtHsp90.6FL was 0.622 Å, while for AsHsp90.6N was 0.545 Å (figure 3*a*). This suggested high structural similarity between the modelled structure and the experimental structure. Secondly, our model for AsHsp90.6N was refined and validated by 1000 ps MD simulations (electronic supplementary material, figure S2), as the Sansom team has previously shown that such short MD simulations were useful to assess the quality of structural models [51].

Sequence alignments (figure 3*b*) suggested residues on L1 and L3 were highly identical to those in all five typical organisms and L1 and L3 can directly interact with ATP. We next conducted 1 μs simulations to analyse the stability of these interactions. Observably, we found the formation of 3–4 contacts between

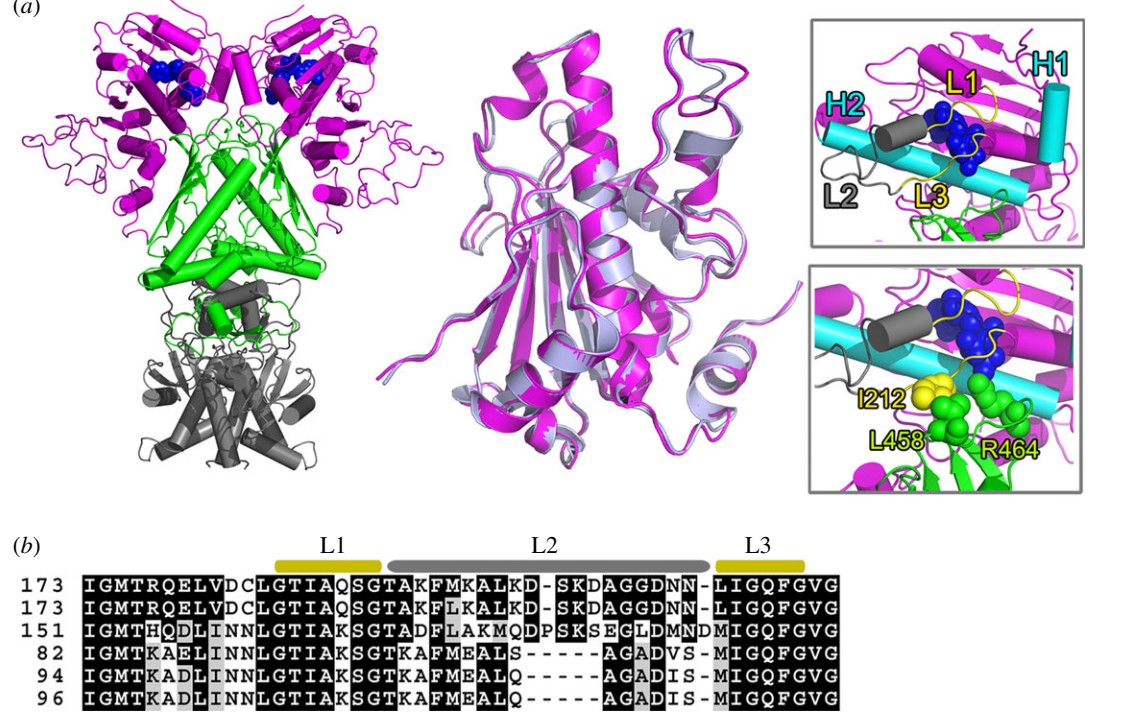

**Figure 3.** The overall structure of AtHsp90.6 and its key domains involved in nucleotide binding. Shown in (*a*) is AtHsp90.6 in cartoon representation (the N-terminal region in pink, the middle region in green and the C-terminal region in grey). The middle panel is superimposition of AtHsp90.6 and the crystal structure (PDB ID: 2CG9). The right panel is a close view of three functional loops, L1, L2 and L3. (*b*) Sequence alignments of AtHsp90.6 in different organisms including *Glycine soja*, *Drosophila melanogaster*, *Saccharomyces cerevisiae*, *Danio rerio* and *Homo sapiens*. Residues on L1, L2 and L3 were highlighted.

residues on L1 and L3 and ATP along the simulations (electronic supplementary material, figure S3). This suggested the ATP-binding property of L1 and L3 was highly conserved among species.

Specifically, residues on L2 showed sequence diversity among species (figure 3*b*). They exhibited high sequence identity in *Glycine soja*, *Drosophila melanogaster*, while low sequence identity in *Saccharomyces cerevisiae*, *Danio rerio* and *Homo sapiens*. This mainly resulted from the various biological functions for L2 and L1/L3. Previous experimental structure [39] suggested two different binding modes of L2 in the ATP- and ADP-bound systems. In the ATP-bound closed state, L2 formed significant interactions with the helix H2, stabilizing ATP in the binding cavity. Supportively, simulations exhibited the number of contacts between L2 and H2 stabilized at 8–10 over the last 100 ns (electronic supplementary material, figure S3). It has been suggested [39] the interaction of L2–H2 was largely disrupted in the ADP-bound open state, and L2 alternatively formed interactions with the helix H1. This suggested the potential function of L2 in the conformational switching between these two states.

The middle domain contained a catalytic loop, consisting of residues D457 to Q468. Simulations suggested residues on this loop can form a significant number of contacts with L3 (electronic supplementary material, figure S3). In particular, L458 forms stably hydrophobic interactions with I212 on L3 (figure 3*a*) and the hydrogen bonding interactions formed between the conserved residue R464 and ATP (figure 3*a*).

## 3.4. The nucleotide binding analysis of AtHsp90.6N

X-ray and Cryo-EM structures revealed the open (o-) and closed (c-) states of Hsp90. To understand the contributions of residues on AtHsp90.6N in these two states, we identified the energy contributions of residues from AtHsp90.6N and compared them with the experimental data.

Figure 4*a* shows the essential residues on AtHsp90.6N contributing to the binding of ATP included N128, C183, L184, S190, G191, G213, Q214 and F215. Their binding energies were −2.61, −1.27, −2.68, −4.65, −4.29, −7.01, −7.79 and −4.68 kcal mol$^{-1}$, respectively. Notably, residues G213, Q214 and F215 were located at the loop L3, while S190 and G191 at L1. This suggested L1 and L3 significantly contributed to

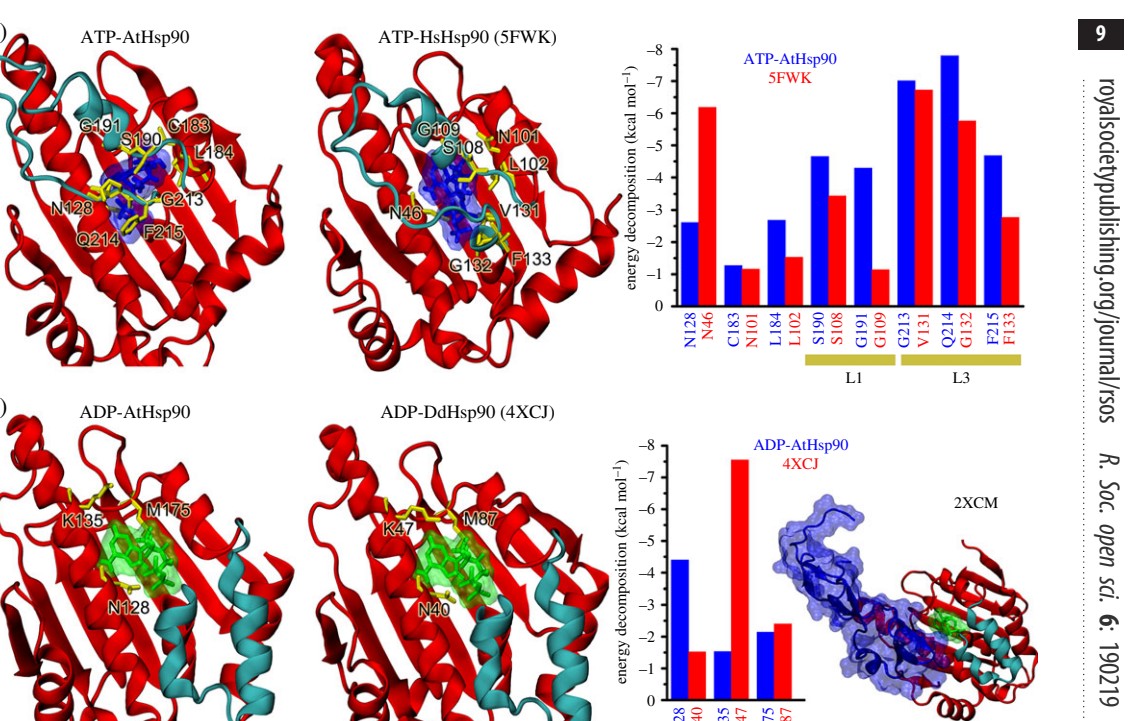

**Figure 4.** Energy decomposition analysis by MM-GBSA. Shown in (*a*) are the key interactions between ATP and Hsp90. The right panel is the per-residue interaction spectrum of Hsp90 with ATP. Shown in (*b*) are the key interactions between ADP and Hsp90. The middle panel is the per-residue interaction spectrum of Hsp90 with ADP. The right panel is the crystal structure of Rar1-ADP-Hsp90.

the binding affinity. In the ADP-bound open state of AtHsp90.6N, three different residues, N128, K135 and M175, contributed −4.4, −1.53 and −2.14 kcal mol$^{-1}$ to the binding energies. The model has a similar binding mode with the crystal structure [38,39], as shown in figure 4*a*,*b* (left and middle panels).

## 3.5. Relative binding free energies in ADP- and ATP-bound AtHsp90.6N

We calculated the relative binding free energies of the complexes ATP- and ADP-bound AtHsp90.6N (table 1). Their energies were −4.47 and 9.90 kcal mol$^{-1}$, suggesting ATP is more favourable to bind AtHsp90.6N while ADP is unfavourable. It has been suggested [52] that Rar participated in the regulation of ATPase activity. We speculated that Rar stayed in the neighbourhood of ADP, acting as a stability enhancer (figure 4*b*).

We decomposed the binding free energies into non-polar and electrostatic components (table 1). Observably, the total non-polar energies contributed to the binding affinity greatly. They included the van der Waals components $\Delta G_{vdw}$ and the solvation non-polar components $\Delta G_{sol-np}$. The total non-polar energies were −58 and −35 kcal mol$^{-1}$ for ATP-AtHsp90.6N and ADP-AtHsp90.6N. Conversely, the total electrostatic interactions exhibited unfavourably contributed to the binding energy. They were 44 and 45 kcal mol$^{-1}$ for ATP-AtHsp90.6N and ADP-AtHsp90.6N. However, structural observations clearly revealed significant interactions such as ATP-Q214. To obtain the explanation, we decomposed the total electrostatic contribution into the gas phase electrostatic $\Delta E_{ele}$ and solvation $\Delta G_{sol,ele}$ components. The gas electrostatic interactions were favourable for the ATP-bound complexes. However, their contributions cannot compensate the desolvation penalties, resulting in the unfavourable electrostatic energies during the binding event.

## 3.6. Large conformational change of L2

Crystallography determined the static closed structure of the ATP-bound Hsp90. We asked if the conformation of Hsp90 changed after the ATP hydrolysis to ADP in the closed state. To this end, we constructed apo-/ATP-/ADP-bound closed AtHsp90.6N, aiming to feature the conformational transitions of AtHsp90.6N.

**Table 1.** The free energies calculated by the MM-GBSA methods for the binding of ADP to its partners DdHsp90 (PDB ID: 4XCJ) and AtHsp90.6, and ATP to its partners HsHsp90 (PDB ID: 5FWK) and AtHsp90.6. $\Delta E_{gas} = \Delta E_{ele} + \Delta E_{vdW}$. $\Delta G_{sol,GB} = \Delta G_{sol\text{-}np} + \Delta G_{GB}$. $\Delta G_{ele\text{-}GB} = \Delta G_{GB} + \Delta E_{ele}$. $\Delta G_{bind,GB} = \Delta E_{gas} + \Delta G_{sol,GB}$.

| contribution | ATP-HsHsp90 | ATP-AtHsp90.6 | ADP-AtHsp90.6 | ADP-DdHsp90 |
|---|---|---|---|---|
| $\Delta E_{ele}$ | $-100.95 \pm 27.63$ | $-151.44 \pm 36.18$ | $248.68 \pm 17.21$ | $250.12 \pm 28.86$ |
| $\Delta E_{vdW}$ | $-54.27 \pm 3.38$ | $-51.90 \pm 2.36$ | $-31.69 \pm 1.92$ | $-31.55 \pm 4.93$ |
| $\Delta E_{gas}$ | $-155.22 \pm 28.21$ | $-203.34 \pm 36.17$ | $217.00 \pm 17.00$ | $218.57 \pm 31.98$ |
| $\Delta G_{GB}$ | $145.43 \pm 27.20$ | $203.51 \pm 34.64$ | $-203.48 \pm 16.37$ | $-213.28 \pm 28.49$ |
| $\Delta G_{sol,np}$ | $-4.60 \pm 0.10$ | $-4.64 \pm 0.10$ | $-3.62 \pm 0.11$ | $-3.53 \pm 0.14$ |
| $\Delta G_{ele\text{-}GB}$ | $44.48 \pm 4.45$ | $52.07 \pm 4.52$ | $45.20 \pm 3.49$ | $36.84 \pm 4.66$ |
| $\Delta G_{bind,GB}$ | $-14.39 \pm 4.84$ | $-4.47 \pm 3.81$ | $9.90 \pm 2.63$ | $1.76 \pm 5.51$ |

Notably, the root mean square fluctuation (RMSF) values on the loop L2 (figure 5a,b) stabilized at approximately 1 Å in ATP-AtHsp90.6N, while it fluctuated around 3.5 Å in ADP-AtHsp90.6N. This emphasized significant local conformational changes of the lid. We next examined the mass distance between the lid and the helix H2 (figure 5c). In ATP-AtHsp90.6N, the distance decreased from 22 Å to 12 Å in the first 40 ns, while stabilizing at 14 Å over the last 50 ns. Similarly, apo-AtHsp90.6 also stabilized around 14 Å from 60 ns to 100 ns. In ADP-AtHsp90.6N, the distance exhibited significant fluctuations along the whole simulations. Supportively, the solvent accessible surface area (SASA) of the lid (figure 5d) fluctuated between 1200 and 1500 Å$^2$ in ADP-AtHsp90.6N, while staying around 1100 Å$^2$ in the ATP-AtHsp90.6N.

## 3.7. Exploring the conformational dynamics by free energy landscape

To further explore the dynamic behaviours of the lid in these two systems, we constructed the free energy landscapes to reflect conformational changes of AtHsp90.6N. We defined two reaction coordinates of the free energy landscapes: one was RMSD of AtHsp90.6N, exhibiting protein structural stability during simulations; the other was the radius of gyrate (Rg) of AtHsp90.6N, reflecting whether AtHsp90.6N was stably folded (figure 6a,b).

Figure 6a,b describes the free energy landscape when AtHsp90.6N is in its apo-state or ATP-bound state. We observed a major well centred at the coordinate (2.3 Å, 16.7 Å) for ATP-AtHsp90.6N, and a major well centred at the coordinate (2.7 Å, 16.9 Å) for apo-AtHsp90.6N. We then extracted snapshots from these two minimum wells and found AtHsp90.6 adopted closed conformations in which the lid closed over the bound ATP. In this closed state, the lid loop L2 stably interacted with the helix H2, while the loops L1 and L3 contributed to the binding of ATP.

Figure 6c describes two major deep wells in the free energy landscape when AtHsp90.6N is bound to ADP. These two wells represented two intermediate states, m1-state and m2-state. The m1-state basin centred at (2.5 Å, 16.9 Å), and the m2-state basin at (3.2 Å, 16.9 Å). These two states were separated by a low barrier of approximately 2.5 $\kappa_B T$.

Extraction of snapshots from these basins suggested significant conformational changes of the lid loop L2 in the intermediate states. We further examined the interactions between residues on L2 and ADP/ATP along the simulations. Figure 7a suggested stable interactions between ATP and six key L2 residues, S202, K203, D204, A205, G206 and G207. However, these interactions were greatly disrupted in ADP-bound simulations.

We then examined the electrostatic potential surfaces by the linearized Poisson–Boltzmann equation mode in the APBS package (figure 7b). In the closed c-state, ATP was embraced by the loop L2 and helix H2, forming the negative electrostatic potentials. In the m2-state, we noticed a significant hole between H2 and L2. The electronegative surface was divided by the electropositive region. However, they still exhibited electrostatic interactions between these two segments, and we speculated that completely disrupting this interaction is a prerequisite for m-state switching to o-state.

# 4. Discussion

Hsp90 is required for correct maturation and activation of client proteins through ATP binding and hydrolysis in the N-terminal domain. It is involved in diverse developmental processes including the

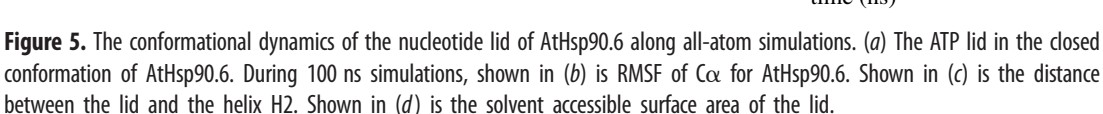

**Figure 5.** The conformational dynamics of the nucleotide lid of AtHsp90.6 along all-atom simulations. (a) The ATP lid in the closed conformation of AtHsp90.6. During 100 ns simulations, shown in (b) is RMSF of Cα for AtHsp90.6. Shown in (c) is the distance between the lid and the helix H2. Shown in (d) is the solvent accessible surface area of the lid.

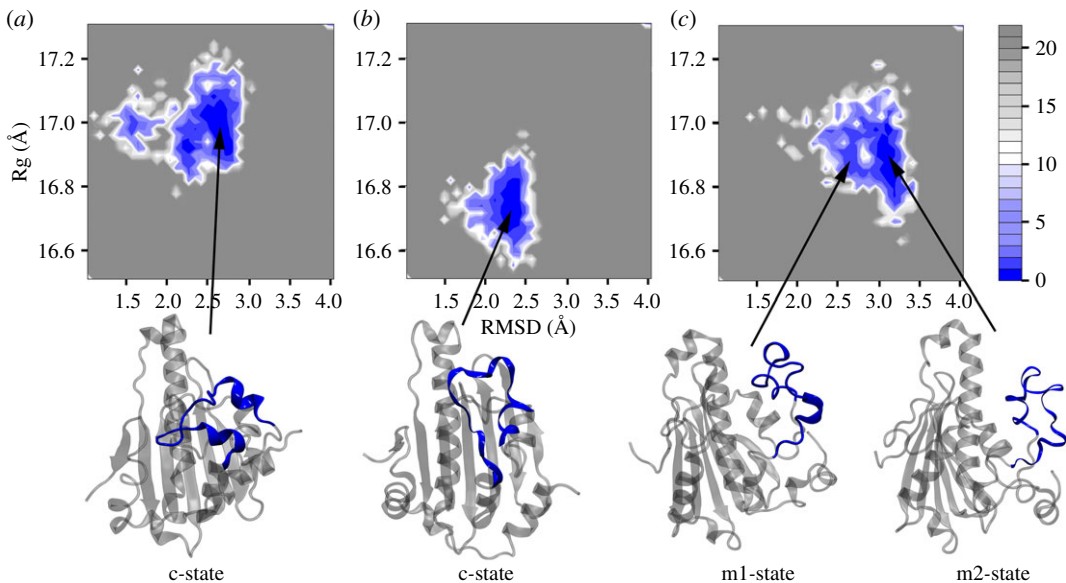

**Figure 6.** The two-dimensional free energy landscape as a function of Rg and RMSD (defined in the text). (a) apo- (b) ATP- and (c) ADP-bound closed conformations of AtHsp90.6 along 100 ns MD simulations. Snapshots from minimum energy wells were extracted.

development of embryo, which is a key component of plant seed vital for human society. In the current study, we have confirmed AtHsp90.6 was essential for plant vitality. Combined with the special expression pattern of AtHsp90.6 in female gametogenesis and early embryogenesis, and the

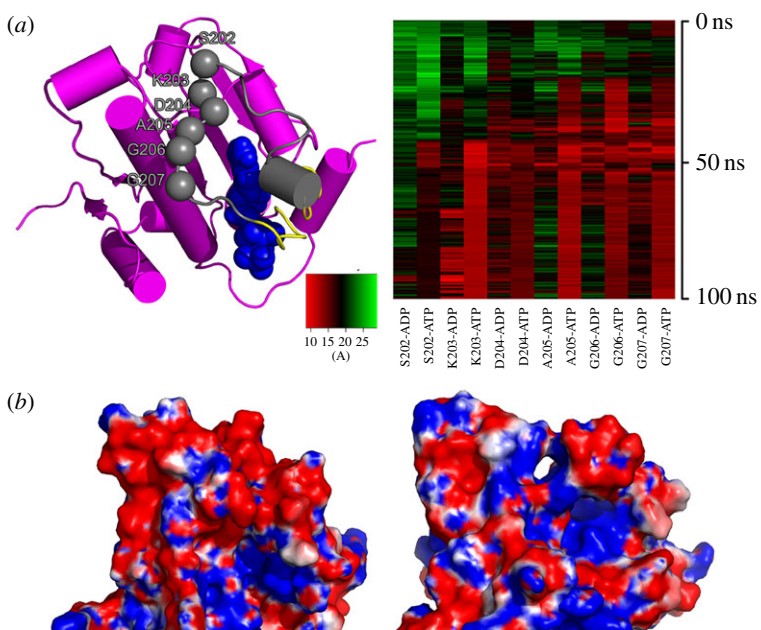

**Figure 7.** The conformational changes of the loop L2 along simulations. (*a*) Interactions between ADP/ATP and residues on the loop L2. (*b*) Skin representation of the electrostatic potential surfaces (ESP) of ATP-/ADP-bound AtHsp90.6 from c-state and m2-state. Red: ESP $< -1 \kappa_B T$ per e, blue: ESP $> -1 \kappa_B T$ per e.

observation that defective embryo in *athsp90.6* mutants terminated at the global-embryo stage (figure 1*l–p*), we proposed that the defect of AtHsp90.6 may lead to congenital dysplasia in early embryos, resulting in the defect of later embryogenesis.

In order to further understanding the crucial protein, we then investigated its structural information. Given experimental attempts to determine the structure of AtHsp90.6 has failed, we used computational approaches to predict the structure of AtHsp90.6 and explore its nucleotide binding mechanism. Our previously studies [14,17,18] highly suggested homology modelling in combination with MD simulations was a useful approach to study the recognition mechanism between ligand and receptor based on available experimental data.

It should be noted that this study identified three functional loops for the N-terminal domain of AtHsp90.6. Structural analysis revealed L1 and L3 contributed significantly to the ATP binding. MM-GBSA also quantitatively supports their importance. This explained previous results from hydrogen exchange mass spectrometry [53] and force constant analysis by simulations [54]. The high sequence similarity of L1 and L3 indicated they adopt a general mechanism among species. This study also identified the loop L2 playing essential roles in the opening and closure of N-terminal lid. This was highly supported by our analysis from RMSF, SAS, free energy landscape and electrostatic potential surfaces. Interestingly, the sequence diversity of L2 among human, yeast and *Arabidopsis* indicated the specificity of L2 in AtHsp90.6.

Pioneering studies [9,38,39] determined the structures of Hsp90 interacting with different partners, providing us invaluable information to observe Hsp90 in its different conformations. The atomic coordinates of HsHsp90 in *H. sapiens* [38] adopted a 'closed' conformation. Five residues, S108, G109, V131, G132 and F133, formed tightly interactions with ATP. These residues were located on the L1 and L3 loops, locking ATP into a stable position. Our study and the resulting structural model supported the 'closed' conformation of AtHsp90.6 when it was in its apo- or ATP-binding state. The atomic coordinates of DdHsp90 in *Dictyostelium discoideum* [39] adopted an 'open' conformation. Three residues, N40, K47 and M87, contributed to the binding of ADP, while all these residues were not on the L1 and L3 loops. This indicated the lid lost the direct interactions with ADP. Interestingly, the atomic coordinates of HvHsp90 in *Hordeum vulgare* [9] suggested the plant Rar1 and Sgt1 can coexist in complexes with Hsp90. Rar1 stabilized the lid of Hsp90 into an open state.

In addition, we asked how the lid of AtHsp90.6 changed from its closed state to its open state. To this end, we exhibited the possibility of their transition by analysing the free energy landscape. We identified the closed states in apo- and ATP-bound AtHsp90.6. Besides, two intermediate states were observed in ADP-bound AtHsp90.6, separated by a barrier of approximately 2.5 $\kappa_B T$. Due to the lack of experimental

measuring the free energy for the transition of the ATP lid on Hsp90, it was still difficult to compare our theoretical data with the experimental data. Our current study should be further explored by the experimental techniques such as single-molecule Förster resonance energy transfer (smFRET). The adenylate kinase (AdK) protein contained an ATP-binding domain (LID) and can transit between an open conformational state and a closed conformational state. In previous simulation work [55], four intermediate states, $\beta$, $\gamma$, $\delta$ and $\varepsilon$, corresponded to the semi-open−semi-closed conformations. Significantly, the intermediates $\gamma$ and $\delta$ had a barrier of approximately $1.02\,\kappa_B T$, $\delta$ and $\varepsilon$ had a barrier of approximately $1.7\,\kappa_B T$, $\beta$ and $\varepsilon$ had a barrier of approximately $2.72\,\kappa_B T$. It should be noted that the protein sequences for ATP lids on AtHsp90.6 and AdK were different. The collective variables and the free energy barriers were also different. Although our current study only provided the prediction for the transition of the ATP lid, we expected the future work would interpret if the transition mechanisms for these two ATP lids were similar or not. We proposed the potential roles of these 'semi-open' states in the releasing step of the Hsp90 client. In the functionally closed state, the lid folded over ATP bound in the pocket. After the client loading, ATP would hydrolysis to ADP and the lid segment swings through nearly '180°', transferring from the closed state to the open state. The tightly bound client-Hsp90 will be released. These multiple 'semi-open' states of ADP-Hsp90 would be beneficial to disrupt the interactions between residues in client-Hsp90 and assist the client releasing.

There were several limitations to the simulation methods used in this paper. One concern was that we only calculated the relative binding free energy between ligands and Hsp90 while entropic contribution to ligand-binding affinity was neglected. Other study groups [45,56] have adopted the relative binding free energy calculation to rank complexes that are closely related. However, the entropy effects play an important role in ligand−receptor interactions and have been increasingly recognized [57]. The binding affinity could be more reliable if the role of entropy were taken into account. Secondly, there were some improved methods for computing the Gibbs free energy of binding reproducibly and accurately [28,58]. Indeed, we previously used hSMD [28] to compute the free-energy profile of phosphatidylserine along its dissociation path of aquaporin 5. Thus, it was essential to determine the Gibbs free energy and the binding mechanics between ligand and Hsp90 by using a variety of improved methods. Thirdly, a variety of experiments such as CASP13 were developed for the assessment of methods of protein structure modelling. In particular, there were various methods to predict the family of Hsp [59−61]. The characteristic of AtHsp90.6 should be further evaluated by these methods. Therefore, we further analysed AtHsp90.6 by using iHSP-PseRAAAC [61] and the results showed it belonged to Hsp family. This correlated well with our previous conclusions.

Data accessibility. The sequence of AtHsp90.6 was obtained from the National Center for Biotechnology Information (NCBI) (Gene ID: 819968) (https://www.ncbi.nlm.nih.gov/gene/). The structure was modelled through the SWISS-MODEL server (https://www.swissmodel.expasy.org/) [19]. The structure of Hsp90 from *Saccharomyces cerevisiae*, *D. discoideum*, *H. sapiens* was obtained from the Protein Data Bank (PDB) (PDB ID: 2CG9, 4XCJ, 5FWK) (https://www.rcsb.org/). Other data supporting this article have been uploaded as a part of the electronic supplementary material.

Authors' contributions. Y.Z. and X.P. planned the experiments. A.L., X.L., Y.Z. and X.Z. performed the experiments and simulations; A.L., X.L., Y.Z., H.Z. and H.D. analysed the results. A.L., X.L., H.D., Y.Z. and X.P. wrote and revised the manuscript. All authors discussed the results and commented on the manuscript. All authors gave final approval for publication.

Competing interests. We declare we have no competing interests.

Funding. The work was financially supported by the National Natural Science Foundation of China (grant nos. 31570317; 31700281; 31700056), and the Plan in Scientific and Technological Innovation Team of Outstanding Young, Hubei Provincial Department of Education (T2017004).

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
