## [Reviewer comments · Royal Society Open Science]

Review History

RSOS-180724.R0 (Original submission)

Review form: Reviewer 1

Is the manuscript scientifically sound in its present form?

Yes

Are the interpretations and conclusions justified by the results?

No

Is the language acceptable?

Yes

Is it clear how to access all supporting data?

Yes

Do you have any ethical concerns with this paper?

No

Have you any concerns about statistical analyses in this paper?

No

Recommendation?

Accept with minor revision (please list in comments)

Comments to the Author(s)

The paper titled "Computational approaches for the study of Hsp90.6 from *Arabidopsis thaliana*" investigated the close and open states by using molecular dynamics. The conformational changes of Hsp90 bound to ATP and ADP were studied. I think the paper has potential to be published. However, several points should be improved.

1. Authors performed two kinds of experiments. However, the correlation between the wet-experiments and dry-experiments were not explained well. From the results and discussion, I cannot understand how to use theory model to explain the biochemical results.
2. The family of Hsp should be further studied by using the methods or webserver provided in references (PMID: 26233307; PMID: 29379521; PMID: 23756733).
3. Authors used Swiss-model to remodel the structure of AtHsp90. I know there are many 3-D structure prediction model. Authors may find them from CASP (<http://predictioncenter.org/casp13/index.cgi>). Many models have displayed very good performance. Why not use them?

Review form: Reviewer 2

Is the manuscript scientifically sound in its present form?

No

Are the interpretations and conclusions justified by the results?

No

Is the language acceptable?

No

Is it clear how to access all supporting data?

Yes

Do you have any ethical concerns with this paper?

No

Have you any concerns about statistical analyses in this paper?

No

Recommendation?

Reject

Comments to the Author(s)

As attached! (Appendix A).

Decision letter (RSOS-180724.R0)

10-Sep-2018

Dear Dr Luo,

Your manuscript ID RSOS-180724 entitled "Computational approaches for the study of Hsp90.6 from *Arabidopsis thaliana*" which you submitted to Royal Society Open Science, has now been reviewed by two independent reviewers. The comments from reviewers are included at the bottom of this letter.

In view of the extensive concerns of the reviewers raised, I regret to inform you that the manuscript has been rejected in its current form. However, a new manuscript may be submitted which takes into consideration all of these comments. Should you decide to resubmit a revised manuscript I would ask to ensure that it is made explicitly clear throughout your manuscript if you are describing an in silico structural analysis or nucleotide binding analysis or an actual wet lab experiment.

Please note that resubmitting your manuscript does not guarantee eventual acceptance, and that your resubmission will be subject to peer review before a decision is made.

Your resubmitted manuscript should be submitted by 10-Mar-2019. If you are unable to submit by this date please contact the Editorial Office.

Please note that Royal Society Open Science will introduce article processing charges for all new submissions received from 1 January 2018. Charges will also apply to papers transferred to Royal Society Open Science from other Royal Society Publishing journals, as well as papers submitted as part of our collaboration with the Royal Society of Chemistry (<http://rsos.royalsocietypublishing.org/chemistry>). If your manuscript is submitted and accepted for publication after 1 Jan 2018, you will be asked to pay the article processing charge, unless you request a waiver and this is approved by Royal Society Publishing. You can find out more about the charges at <http://rsos.royalsocietypublishing.org/page/charges>. Should you have any queries, please contact openscience@royalsociety.org.

on behalf of Professor Diwakar Shukla (Associate Editor) and Professor Katrin Rittinger (Subject Editor)

Reviewers' Comments to Author:

Reviewer: 1

Comments to the Author(s)

The paper titled "Computational approaches for the study of Hsp90.6 from *Arabidopsis thaliana*" investigated the close and open states by using molecular dynamics. The conformational changes of Hsp90 bound to ATP and ADP were studied. I think the paper has potential to be published. However, several points should be improved.

1. Authors performed two kinds of experiments. However, the correlation between the wet-experiments and dry-experiments were not explained well. From the results and discussion, I cannot understand how to use theory model to explain the biochemical results.
2. The family of Hsp should be further studied by using the methods or webservers provided in references (PMID: 26233307; PMID: 29379521; PMID: 23756733).
3. Authors used Swiss-model to remodel the structure of AtHsp90. I know there are many 3-D structure prediction model. Authors may find them from CASP (<http://predictioncenter.org/casp13/index.cgi>). Many models have displayed very good performance. Why not use them?

Reviewer: 2

Comments to the Author(s)

As attached!

Author's Response to Decision Letter for (RSOS-180724.R0)

See Appendix B.

RSOS-190219.R0

Review form: Reviewer 1

Is the manuscript scientifically sound in its present form?

Yes

Are the interpretations and conclusions justified by the results?

Yes

Is the language acceptable?

Yes

Is it clear how to access all supporting data?

Yes

Do you have any ethical concerns with this paper?

No

Have you any concerns about statistical analyses in this paper?

Yes

Recommendation?

Accept as is

Comments to the Author(s)

Authors have answered my questions. I think the paper can be accepted.

Review form: Reviewer 2

Is the manuscript scientifically sound in its present form?

Yes

Are the interpretations and conclusions justified by the results?

Yes

Is the language acceptable?

Yes

Is it clear how to access all supporting data?

Yes

Do you have any ethical concerns with this paper?

No

Have you any concerns about statistical analyses in this paper?

No

Recommendation?

Accept with minor revision (please list in comments)

Comments to the Author(s)

Included in the attached file (Appendix C).

Decision letter (RSOS-190219.R0)

21-Mar-2019

Dear Dr Iuo,

On behalf of the Editor, I am pleased to inform you that your Manuscript RSOS-190219 entitled "Identification of AtHsp90.6 involved in early embryogenesis and its structure prediction by molecular dynamics simulations" has been accepted for publication in Royal Society Open Science subject to minor revision in accordance with the referee suggestions. Please find the referees' comments at the end of this email.

The reviewers and Subject Editor have recommended publication, but also suggest some minor revisions to your manuscript. Therefore, I invite you to respond to the comments and revise your manuscript.

- Ethics statement

- Data accessibility

If you wish to submit your supporting data or code to Dryad (<http://datadryad.org/>), or modify your current submission to dryad, please use the following link:
<http://datadryad.org/submit?journalID=RSOS&manu=RSOS-190219>

- Competing interests

- Authors' contributions

AB carried out the molecular lab work, participated in data analysis, carried out sequence alignments, participated in the design of the study and drafted the manuscript; CD carried out

the statistical analyses; EF collected field data; GH conceived of the study, designed the study, coordinated the study and helped draft the manuscript. All authors gave final approval for publication.

- Acknowledgements

- Funding statement

Because the schedule for publication is very tight, it is a condition of publication that you submit the revised version of your manuscript before 30-Mar-2019. Please note that the revision deadline will expire at 00.00am on this date. If you do not think you will be able to meet this date please let me know immediately.

Supplementary files will be published alongside the paper on the journal website and posted on the online figshare repository (<https://figshare.com>). The heading and legend provided for each supplementary file during the submission process will be used to create the figshare page, so

please ensure these are accurate and informative so that your files can be found in searches. Files on figshare will be made available approximately one week before the accompanying article so that the supplementary material can be attributed a unique DOI.

on behalf of Professor Diwakar Shukla (Associate Editor) and Professor Katrin Rittinger (Subject Editor)
openscience@royalsociety.org

Reviewer comments to Author:
Reviewer: 1

Comments to the Author(s)
Authors have answered my questions. I think the paper can be accepted.

Reviewer: 2

Comments to the Author(s)
Included in the attached file

Author's Response to Decision Letter for (RSOS-190219.R0)

See Appendix D.

Decision letter (RSOS-190219.R1)

02-Apr-2019

Dear Dr Luo,

I am pleased to inform you that your manuscript entitled "Identification of AtHsp90.6 involved in early embryogenesis and its structure prediction by molecular dynamics simulations" is now accepted for publication in Royal Society Open Science.

You can expect to receive a proof of your article in the near future. Please contact the editorial office (openscience_proofs@royalsociety.org and openscience@royalsociety.org) to let us know if

you are likely to be away from e-mail contact. Due to rapid publication and an extremely tight schedule, if comments are not received, your paper may experience a delay in publication.

on behalf of Professor Diwakar Shukla (Associate Editor) and Katrin Rittinger (Subject Editor)
openscience@royalsociety.org

Appendix A

This manuscript describes the exploration of the nucleotide binding mechanism in the closed “lid-down” and open “lid-up” states by molecular dynamics simulations. Free energy landscape and electrostatic potential analysis were used to address the switching mechanism between the closed and open states. The manuscript mainly showed results, but the discussion was not deep enough, not many comparisons were done to previous studies. Thus, I feel that it is not suitable for publishing in Royal Society Open Science. To improve this manuscript, I would like to make the following comments:

1. From the title, it seems to be a review on computational approaches for the study of Hsp90.6. There are some experiments, but I did not see how the experimental results are related to the results in silicon.

2. The authors used homology modeling to generate the models for AtHsp90.6 with SWISS-MODEL, the crystal structure PDB ID 2CG9 was used as a template. However, the sequence identity was not reported, the sequence alignment was shown, the number of generated models was not mentioned, the assessment of the generated models and how the models were selected were not discussed.

3. In the Material and Methods section, there are three types of molecular dynamics simulations, 1) 100 ns MD simulations using OPLS force field and spc216 water model, performed with Gromacs; 2) 1000 ns CG MD simulations using MARTINI force field, performed with Gromacs; 3) MD simulations using AMBER ff99 force field for MM-GBSA calculations, performed with Amber 11.

a. It is not very clear how many systems were simulated from the Methods section, the readers can only guess from the results sections. From Figure 5 and 6, I speculate that two systems were simulated for 100 ns, respectively, closed ATP-AtHsp90.6 and closed ADP-AtHsp90.6. Why the Apo AtHsp90.6 at two different states were not simulated as the authors also wanted to study the binding of ATP? How many systems were simulated with CG model? I suspect only ATP-AtHsp90.6 was simulated, but not ADP-AtHsp90.6, then why not?

Section 3.7 (line 29), the authors mentioned “The initial structure of N-terminal AtHsp90.6 was immersed...”, did the authors only simulate the N-terminal of AtHsp90.6 or the whole structure of AtHsp90.6, but it was mentioned that the full length of AtHsp90.6 was modeled in Section 3.6? This is confusing.

b. Three force fields were used in this study, how could the authors justify and correlate the results from three different parameter sets? why not just use one? And I did not see the purpose of running CG MD simulations. The authors even did not mention the simulation time with Amber 11, and how many configurations were used for the MM-GBSA calculations.

c. The parameters of ADP and ATP for the three force fields were not mentioned in the Methods sections, the authors developed, or used existed parameters?

d. Section 3.9 (page 9), the authors stated “Because of the constant contribution of $-T\Delta S$ for each complex...”, this is not correct, the entropy is not constant, and it can be estimated with normal mode analysis, as done in this paper J. Med. Chem. 2006,49,6596-6606.

4. In Figure 3C, the authors showed the number of contacts of L1-ATP, L3-ATP, L2-H2 and L3-catalytic loop, it is not clear how the number of contacts were calculated, what's the cut-off? The y-axis ranges of the four panels are different, this does not look obvious.

5. Section 4.4 (line 52-53), “These values were in good agreement with the experimental data[32] (Figure 4A)”, “This also correlated well with the experimental data [31] (Figure 4B)”, I don't see how these values can be in agreement with a crystal structure, one can only state that the model have a

similar binding mode with the crystal structure, as shown in Figure 4A and 4B (left and middle panels). From the right panel of Figure 4A and 4B, Table 1, I see that MD simulations were also performed for the structures PDB IDs 5FWK and 4XCJ, and then MM/GBSA calculations were done, but the simulations were not mentioned in the Method section and these results were not discussed fully in the main text. The authors did not explain why those simulations were performed. Why did the authors show the structure of 2XCM in Figure 4B?

6. In Figure 5, the distance was shown for all the 100 ns, but the SASA was only shown from 70 to 100 ns, why?

7. In Figure 6B, two intermediates were found, and they were separated by a barrier of ~ 2.5 $k_B T$. Does it agree with the experimental data?

8. Fig was used in the main text while Figure was used in the caption. They should be consistent with each other.

9. There were a lot of missing references:

a. OPLS force field, SPC216 water model, PME algorithm, MARTINI force field, Berendsen thermostat and barostat methods, AMBER ff99 force field, Amber 11 package, SPSS19 software,

b. MM/GBSA, the reference [26] given was wrong, which is for GB implicit water model, but not for MM/GBSA.

c. Protein Data Bank, PDB ID 2CG9, 5FWK, 4XCJ, 2XCM, Blast algorithm

d. Page 15, line 3, “using a GB surface area (GBSA) method⁵⁴”, the total number of references is 35, I don't know how [54] is coming from.

10. a. There are missing supplementary figure and table: Figure S1 and Table S1 as mentioned in Section 4.1 ;

b. Page 11, “no only forms” should be “not only forms” ;

c. There should be a blank between digit and units, “1000ns” should be “1000 ns” ... ;

d. 1000 kJ mol⁻¹ nm⁻², superscript should be used ;

e. “using a steepest descent method”, “a” should be removed ;

f. “Conformation dynamics” should be “conformational dynamics” in the Summary, “conformation changes” on page 19 line 46 should be “conformational changes”

g. Abbreviations should be consistent, athsp90.6 should be AtHsp90.6, there are also some abbreviations with full names, ABRC, TAIL-PCR, CLSM, Rg, GFP

Appendix B

Dear reviewers,

Thanks for your comments on our paper. Your comments were highly insightful and enabled us to greatly improve the quality of our manuscript.

In the following pages are our point-by-point responses to each of the comments of the reviewers. Revisions in the text are shown using red color for additions, and blue strikethrough font for deletions. In order to account for the reviewer #2's suggestions to adding simulation systems for apo-AtHsp90.6, we now run independent simulations. As a consequence, we entirely reconstructed corresponding parts from the "result" and "discussion" section.

We hope that these revisions in the manuscript and our accompanying responses will be sufficient to make our manuscript suitable for publication in *Royal Society Open Science*.

Yours sincerely,

Dr. Yubo ZHANG

Department of Food Science, Foshan University, Foshan, P.R.China

Email: zhyubo7@gmail.com

Prof. Xiongbo PENG

College of Life Science, State Key Laboratory of Hybrid Rice, Wuhan University, Wuhan, P.R.China

Email: Bobopx@whu.edu.cn

Responses to the comments of **Reviewer #1**

Comments 1. Authors performed two kinds of experiments. However, the correlation between the wet-experiments and dry-experiments were not explained well. From the results and discussion, I cannot understand how to use theory model to explain the biochemical results.

Response: Thank you for pointing out.

In the wet experiments, we identified AtHsp90.6 was essential for early embryogenesis and plant vitality for the first time by screening out a T-DNA insertion *athsp90.6* mutant from ~7600 mutants. Since a so important protein is found, we are interested in its structural information for further understanding this protein.

Given experimental attempts to determine the structure of AtHsp90.6 has failed, we used computational approaches to predict conformational dynamics and nucleotide binding mechanism of AtHsp90.6. We identified three functional loops for the N-terminal domain of AtHsp90.6 and observed two intermediate states in ADP-bound AtHsp90.6. We hope these computational studies would be helpful for protein structure prediction.

Comments 2. The family of Hsp should be further studied by using the methods or webservers provided in references (PMID: 26233307; PMID: 29379521; PMID: 23756733).

Response: We agree. We now re-evaluated the characteristic of AtHsp90.6 by using iHSP-PseRAAC provided in the reference (PMID: 23756733). The results showed our sequence is Hsp90, correlating well with our previous conclusion.

Comments 3. Authors used Swiss-model to remodel the structure of AtHsp90. I know there are many 3-D structure prediction models. Authors may find them from CASP (<http://predictioncenter.org/casp13/index.cgi>). Many models have displayed very good performance. Why not use them?

Response: Thank you for suggestions. Given the sequence identity between our model and the crystal structure (PDB ID 2CG9) was 56%, we homology modelled AtHsp90.6 by our established methods (PMID: 20936826, 21637806 and 22653607). To assess the generated model, we adopted two criteria:

Firstly, the structural similarity was measured by RMSD in their best-superimposed atomic coordinates. The RMSD is 0.545 Å, suggesting highly structural similarity between our model and the experimental structure.

Secondly, our model was refined and validated by 1000 ps MD simulations [Fig. S2], as the Sansom team has previously shown that such short MD simulations are useful to assess the quality of structural models (PMID: 16102990).

It is a good choice to predict model from CASP and necessary discussion has been made in the revised manuscript.

Again, we would like to thank the reviewer #1 for his/her insights and constructive suggestions!

Responses to the comments of Reviewer #2

Major revision

Comments 1. From the title, it seems to be a review on computational approaches for the study of Hsp90.6. There are some experiments, but I did not see how the experimental results are related to the results in silicon.

Response: Thank you for pointing out.

In the wet experiment, we identified AtHsp90.6 was essential for early embryogenesis and plant vitality for the first time by screening out a T-DNA insertion *athsp90.6* mutant from ~7600 mutants. Since a so important protein is found, we are interested in its structural information for further understanding this protein.

Given experimental attempts to determine the structure of AtHsp90.6 has failed, we used computational approaches to predict the structure of AtHsp90.6 and explore its conformational dynamics and nucleotide binding mechanism. Firstly, we identified three functional loops for the N-terminal domain of AtHsp90.6. Secondly, we observed the closed states in apo- and ATP-bound AtHsp90.6. Two intermediate states were observed in ADP-bound AtHsp90.6, separated by a barrier of ~2.5 kBT. We hope these computational studies would be helpful for protein structure prediction.

To better describe wet and dry experiments in the current study, we would like to change the title to “Identification of AtHsp90.6 involved in early embryogenesis and its structure prediction by molecular dynamics simulations”. Necessary discussion has been made in the revised manuscript.

Comments 2. The authors used homology modeling to generate the models for AtHsp90.6 with SWISS-MODEL, the crystal structure PDB ID 2CG9 was used as a template. However, the **sequence identity** was not reported, the sequences alignment was shown, the number of generated

models was not mentioned, **the assessment of the generated models** and how the models were selected were not discussed.

Response: We agree sequence identity and model assessment were not well explained.

1) The sequence identity between our model and the crystal structure PDB ID 2CG9 was 56%, and this was now shown in the methods section.

2) We modeled AtHsp90.6 by using the crystal structure from *Saccharomyces cerevisiae* (PDB id: 2CG9, sequence identity 56%) as a template through the SWISS-MODEL server by our established methods (PMID: 20936826, 21637806 and 22653607). SWISS-MODEL is a web-based integrated service dedicated to protein structure homology and we used the default parameters to generate the model.

To assess the generated model, we adopted two criteria:

Firstly, the structural similarity between our model and the crystal structure was measured by the root-mean-square-deviation (RMSD) in their best-superimposed atomic coordinates. The RMSD is 0.545 Å, suggesting high structural similarity between these two structures.

Secondly, our model was refined and validated by 1000 ps MD simulations [Fig. S2], as the Sansom team has previously shown that such short MD simulations are useful to assess the quality of structural models (PMID: 16102990).

Comments 3. In the Material and Methods section, there are three types of molecular dynamics simulations, 1) 100 ns MD simulations using OPLS force field and spc216 water model, performed with Gromacs; 2) 1000 ns CG MD simulations using MARTINI force field, performed with Gromacs; 3) MD simulations using AMBER ff99 force field for MM-GBSA calculations, performed with Amber 11.

a. It is not very clear **how many systems were simulated** from the Methods section, the readers can only guess from the results sections. From Figure 5 and 6, I speculate that two systems were simulated for 100 ns, respectively, closed ATP-AtHsp90.6 and closed ADP-AtHsp90.6.

Response: We now summarized our simulation systems, shown as Table S2.

Comments 4. Why the Apo AtHsp90.6 at two different states were not simulated as the authors also wanted to study the binding of ATP?

Response: Thank you for pointing out.

To provide more quantitative statistic analysis, we now ran three independent 100ns explicit-solvent MD simulations with AMBER ff99 force fields and the TIP3P water model. We analyzed the results in the following parts.

1) We re-evaluated the RMSF of C α for AtHsp90.6, the distance between the lid and the helix H2, and the solvent accessible surface area of the lid. Our new, extended analysis confirms the conformational changes of the ATP lid, and we have fully revised the corresponding analysis.

2) We re-analyzed the two-dimensional free energy landscape as a function of R g and RMSD of AtHsp90.6. We found a good agreement between our computational data and previous simulation work of the ATP lid on AdK.

Comments 5. How many systems were simulated with CG model? I suspect only ATP-AtHsp90.6 was simulated, but not ADP-AtHsp90.6, then why not? Section 3.7 (line 29), the authors mentioned “The initial structure of N-terminal AtHsp90.6 was immersed...”, did the authors only

simulated the N-terminal of AtHsp90.6 or the whole structure of AtHsp90.6, but it was mentioned that the full length of AtHsp90.6 was modeled in Section 3.6? This is confusing.

Response: Thank you for pointing out.

1) Given the lower-resolution coarse-grained model, we only used CG MD simulations to preliminarily test if key contacts can stabilize during long simulations. To study the conformational changes, we used 100 ns MD simulations using AMBER ff99 force fields, performed with Gromacs shown on Table S2. To make the article coherence, we now put the results of CG MD simulations to supplementary materials.

2) We modelled the whole structure of AtHsp90.6 based on the crystal structure (PDB id: 2CG9, sequence identity 56%) as a template. This whole structure was named as AtHsp90.6FL shown on Table S2. Given the nucleotide can directly bind to the N-terminal of AtHsp90.6, we select the N-terminal for the nucleotide binding study and conformation dynamics study. The N-terminal of AtHsp90.6 was now named as AtHsp90.6N.

Comments 6.

b. Three force fields were used in this study, how could the authors justify and correlate the results from three different parameter sets? **why not just use one?** And I did not see the purpose of running CG MD simulations. The authors even did not mention the simulation time with Amber 11, and how many configurations were used for the MM-GBSA calculations.

Response: Thank you for pointing out.

1) We now ran three independent 100 ns explicit-solvent MD simulations with AMBER ff99 force fields and have fully revised the corresponding analysis.

2) To make the article coherence, we now put the results of CG MD simulations to supplementary materials.

3) Given previous MM-GBSA studies (PMID: 22339124, 23595060, 24683369) showed 30, 40 and 100 snapshots and short simulation time could give reasonable results in correlation with experimental data, we conducted 1000 ps simulation time for MM-GBSA and collected 50 snapshots at the final equilibrium stage for calculation (shown on Table S2 and materials).

Comments 7.

c. The parameters of ADP and ATP for the three force fields were not mentioned in the Methods sections, the authors developed, or used existed parameters?

Response: We agree the parameters should be mentioned in the Methods sections.

1) The parameters for ADP and ATP were taken from the AMBER parameter database, maintained by The Bryce Group (<http://research.bmh.manchester.ac.uk/bryce/amber>). The parameters were developed by Carlson HA (PMID: 12759902), and their details have been put in the Supplemental materials.

2) To perform MD simulations with the GROMACS software package, we conduct the conversion to GROMACS compatible topology using ACPYPE (PMID: 22824207).

Comments 8.

d. Section 3.9 (page 9), the authors stated “Because of the constant contribution of $-T\Delta S$ for each complex...”, this is not correct, the entropy is not constant, and it can be estimated with normal mode analysis, as done in this paper J. Med. Chem. 2006,49,6596-6606.

Response: We agree with the reviewer. The sentences as mentioned in Section 3.9 have been removed completely. We agree entropy effects play an important role in ligand-receptor interactions. Especially, our improved hSMD method (PMID: 23176748) would provide more efficiency in free energy calculation. In this study, we neglected the entropic and focused on the relative binding free energy to rank complexes that are closely related as suggested by (PMID 20936826, 26507522). We are totally open, and would like to seek for more tools for free energy analysis, as publication is clearly not the ultimate goal of our endeavor.

Comments 9. In Figure 3C, the authors showed the number of contacts of L1-ATP, L3-ATP, L2-H2 and L3- catalytic loop, it is not clear how the number of contacts were calculated, what's the cut-off? The y-axis ranges of the four panels are different, this does not look obvious.

Response: We agree. Necessary information have been added in the revised manuscript, where the cut-off value of the number of contacts were 0.7 nm according to Sansom's method (PMID: 20409475, 24204243) and y-axis ranges of the four panels are same now.

Comments 10. Section 4.4 (line 52-53), “These values were in good agreement with the experimental data [32] (Figure 4A)”, “This also correlated well with the experimental data [31] (Figure 4B)”, I don't see how these values can be in agreement with a crystal structure, one can only state that the model have a similar binding mode with the crystal structure, as shown in Figure 4A and 4B (left and middle panels).

Response: We agree. Necessary changes in these statements have been made in the Section 4.4(line 52-53).

Comments 11. From the right panel of Figure 4A and 4B, Table 1, I see that MD simulations were also performed for the structures PDB IDs 5FWK and 4XCJ, and then MM/GBSA calculations were done, but the simulations were not mentioned in the Method section and these results were not discussed fully in the main text. The authors did not explain why those simulations were performed. Why did the authors show the structure of 2XCM in Figure 4B?

Response: We agree our missing necessary explanations for crystal structures. We reconstruct these sections as following:

- 1) Simulation details for the structures (PDB code: 5FWK and 4XCJ) have been put in the Methods section of the revised manuscript.
- 2) The atomic coordinates of HsHsp90 (PDB code: 5FWK) in *H.sapiens* adopted a “closed” conformation while the atomic coordinates of DdHsp90 (PDB code: 4XCJ) in *D.discoideum* adopted an “open” conformation. We performed MD simulations for these two experimental structures to understand if the “open” and “closed” conformations of AtHsp90.6 were in good agreement with the experimental results. We now added this information in the revised manuscript.
- 3) The atomic coordinates of HvHsp90 (PDB code: 2XCM) in *H.vulgare* represented a particular “open” conformation of Hsp90, while Rar1 stabilized the lid of Hsp90 into an open state. We now added this information in the revised manuscript.

Comments 12. In Figure 5, the distance was shown for all the 100 ns, but the SASA was only shown from 70 to 100 ns, why?

Response: We agree. We had now shown the SASA from 0 to 100 ns.

Comments 13. In Figure 6B, two intermediates were found, and they were separated by a barrier of ~2.5 kBT. Does it agree with the experimental data?

Response: Our data correlates well with previously published data of adenylate kinase (AdK) protein (PMID: 26244746). In the current study, two intermediate states were observed in ADP-bound AtHsp90.6, separated by a barrier of ~2.5 kBT. Interestingly, AdK contained an ATP-binding domain (LID), can transit between an open conformational state and a closed conformational state. Four intermediate states β , γ , δ and ϵ corresponded to the semi-open–semi-closed conformations. The intermediates γ and δ had a barrier of ~1.02 kBT, the intermediates δ and ϵ had a barrier of ~1.7 kBT while the intermediates β and ϵ had a barrier of ~2.72 kBT. Thus, our data agrees well with this published data.

Comments 14. Fig was used in the main text while Figure was used in the caption. They should be consistent with each other.

Response: Thank you for pointing out. The term “Fig” used in the main text has been changed to “figure” completely.

Minor revision

Comments 15. There were a lot of missing references:

a. OPLS force field, SPC216 water model, PME algorithm [1], MARTINI force field [2], Berendsen thermostat and barostat methods [3], AMBER ff99 force field [4], Amber 11 package [5], SPSS19 software [6]

Response: Missing references had been added in the revised manuscript.

b. MM/GBSA, the reference [26] given was wrong, which is for GB implicit water model, but not for MM/GBSA.

Response: References had been corrected for MM/GBSA.

c. Protein Data Bank, PDB ID 2CG9, 5FWK, 4XCJ, 2XCM, Blast algorithm

Response: Missing references had been added in the revised manuscript.

d. Page 15, line 3, “using a GB surface area (GBSA) method 54”, the total number of references is 35, I don't know how [54] is coming from.

Response: Reference [54] has been removed completely in the revised manuscript.

Comments 16.

a. There are missing supplementary figure and table: Figure S1 and Table S1 as mentioned in Section 4.1;

Response: Done in the electronic supplementary material.

b. Page 11, “no only forms” should be “not only forms” ;

Response: Done in the revised Page 11.

c. There should be a blank between digit and units, “1000ns” should be “1000 ns” ... ;

Response: Done in the revised manuscript.

d. 1000 kJ mol⁻¹ nm⁻², superscript should be used;

Response: We now changed “1000 kJ mol⁻¹ nm⁻²” to “1000 kJ mol⁻¹ nm⁻²” completely.

e. “using a steepest descent method”, “a” should be removed;

Response: We now remove “a” from “using a steepest descent method” completely.

f. “Conformation dynamics” should be “conformational dynamics” in the Summary; “conformation changes” on page 19 line 46 should be “conformational changes”

Response: Done in the Summary and page 19 line 46.

g. Abbreviations should be consistent, athsp90.6 should be AtHsp90.6, there are also some abbreviations with full names, ABRC, TAIL-PCR, CLSM, Rg, GFP

Response: We now revised abbreviations “ABRC, TAIL-PCR, CLSM, Rg, GFP” with full names completely. In this work, we continued to use our previously writing format (PMID: 23085019, 21889052 and 24996653), in which the mutant (*athsp90.6*) is in italic lowercase letters format, the gene (*AtHsp90.6*) is in italic uppercase letters format, the protein (AtHsp90.6) is in normal uppercase letters format.

Again, we would like to thank the reviewer #2 for his/her detailed comments and suggestions for the manuscript.

Appendix C

A major revision was done on this manuscript, new simulations and new analyses were performed, and new results were included. However, there is still one major issue that I am concerned.

For the reply to Comment 13 of Reviewer 2:

“ Comments 13. In Figure 6B, two intermediates were found, and they were separated by a barrier of ~ 2.5 kBT. Does it agree with the experimental data?

Our data correlates well with previously published data of adenylate kinase (AdK) protein (PMID: 26244746). In the current study, two intermediate states were observed in ADP-bound AtHsp90.6, separated by a barrier of ~ 2.5 kBT. Interestingly, AdK contained an ATP-binding domain (LID), can transit between an open conformational state and a closed conformational state. Four intermediate states β , γ , δ and ϵ corresponded to the semi-open–semi-closed conformations. The intermediates γ and δ had a barrier of ~ 1.02 kBT, the intermediates δ and ϵ had a barrier of ~ 1.7 kBT while the intermediates β and ϵ had a barrier of ~ 2.72 kBT. Thus, our data agrees well with this published data. ”

In Figure 6, three states were found in this study, and the energy differences between states were compared in a previous computational study, reference [54]. However, the agreement between the current and previous studies is not convincing. The proteins are different, three states were found in this work, while 4 states were found in the previous study. And the collective variables used for the 2D free energy landscapes, and the free energy barriers between states are also different (Figure 6 of reference [54]). Are the conformations of the states found in this work similar to the states in the previous study?

Another thing, the reviewer asked for experimental data, while the authors cited computational work, which is not answering the reviewer directly.

And there are still some minor issues:

1. The coarse grained parameters of ATP were still not explained in the Methods section, reference is needed for it.
2. page 6, $(\text{kcal/mol})/\text{\AA}^2$ should be $\text{kcal mol}^{-1} \text{\AA}^{-2}$
3. In the main text, when a figure is referred, figure X should be Figure X.
4. In Figure S2, the unit of RMSD should be angstrom, but not “A”, though we can use “A” for angstrom in informal documents.

Appendix D

Dear reviewers,

Thanks for your comments on our paper. Your comments were highly insightful and enabled us to greatly improve the quality of our manuscript.

We thank Reviewer #1 for his/her endorsement. In the following pages are our point-by-point responses to each of the comments of the Reviewer #2. Revisions in the text are shown using red color for additions, and blue strikethrough font for deletions.

We hope that these revisions in the manuscript and our accompanying responses will be sufficient to make our manuscript suitable for publication in *Royal Society Open Science*.

Yours sincerely,

Dr. Yubo ZHANG

Department of Food Science, Foshan University, Foshan, P.R.China

Email: zhyubo7@gmail.com

Prof. Xiongbo PENG

College of Life Science, State Key Laboratory of Hybrid Rice, Wuhan University, Wuhan, P.R.China

Email: Bobopx@whu.edu.cn

Responses to the comments of **Reviewer #2**

Comments.

A major revision was done on this manuscript, new simulations and new analyses were performed, and new results were included. However, there is still one major issue that I am concerned.

For the reply to Comment 13 of Reviewer 2:

“Comments 13. In Figure 6B, two intermediates were found, and they were separated by a barrier of ~2.5 kBT. Does it agree with the experimental data?”

Our data correlates well with previously published data of adenylate kinase (AdK) protein (PMID: 26244746). In the current study, two intermediate states were observed in ADP-bound AtHsp90.6, separated by a barrier of ~2.5 kBT. Interestingly, AdK contained an ATP-binding domain (LID), can transit between an open conformational state and a closed conformational state. Four intermediate states β , γ , δ and ϵ corresponded to the semi-open–semi-closed conformations. The intermediates γ and δ had a barrier of ~1.02 kBT, the intermediates δ and ϵ had a barrier of ~1.7 kBT while the intermediates β and ϵ had a barrier of ~2.72 kBT. Thus, our data agrees well with this published data.”

In Figure 6, three states were found in this study, and the energy differences between states were

compared in a previous computational study, reference [54]. However, the agreement between the current and previous studies is not convincing. **The proteins are different**, three states were found in this work, while 4 states were found in the previous study. And the **collective variables** used for the 2D free energy landscapes, and the **free energy barriers** between states are also different (Figure 6 of reference [54]). **Are the conformations of the states found in this work similar to the states in the previous study?**

Another thing, the reviewer asked for **experimental data**, while the authors cited computational work, which is not answering the reviewer directly.

Response: Thank you for pointing out. We have indeed tried very hard to find out experimental data to compare with our computational data. However, we did not have any success. In reference [54](in the revised manuscript was [55]), Li et al. found the intermediate states had barriers from 1.02 to 2.72 kBT by using collective variables LID-CORE and NMP-CORE angles. We examined the conformations of AtHsp90.6. Although AtHsp90.6's ATP lid can form open, intermediate, and closed conformations, we should admit the sequences of these two ATP lids are different. We have now removed "Our data correlates well with previously published data of adenylate kinase (AdK) protein" completely. In the revision, we explained the difference of the protein in our work compared with that in the reference [54] (in the revised manuscript was [55]). In particular, we pointed out this study can only provide the prediction for the energy barrier between different conformations of AtHsp90.6. The accurate dynamics information of the ATP lid on AtHsp90.6 needs to be described by powerful experimental techniques such as single molecule Förster Resonance Energy Transfer.

And there are still some minor issues:

1. The coarse grained parameters of ATP were still not explained in the Methods section, reference is needed for it.

Response: The coarse grained parameters of ATP were taken from Ref. [PMID: 26574472] with minor modifications. Missing reference had been added in the revised manuscript. We also attached the details in the following:

; Topology requires martini_v2.1-dna.itp

[moleculetype]

; Name Exclusions
 ATP 1

[atoms]

1	Qa	1	ATP	PG	1	-2.0000 ; ATP
2	Qa	1	ATP	PB	2	-1.0000 ; ATP
3	Qa	1	ATP	PA	3	-1.0000 ; ATP
4	SN0	1	ATP	RB1	4	0.0000 ; ATP
5	P3	1	ATP	RB2	5	0.0000 ; ATP
6	TN0	1	ATP	SC1	6	0.0000 ; ATP
7	TN0	1	ATP	SC2	7	0.0000 ; ATP
8	TP1	1	ATP	SC3	8	0.0000 ; ATP
9	TNa	1	ATP	SC4	9	0.0000 ; ATP

[bonds]

; Backbone bonds

1	2	1	0.27900	75000 ; ATP(PG)-ATP(PB)
2	3	1	0.27900	75000 ; ATP(PB)-ATP(PA)
3	4	1	0.35600	20000 ; ATP(PA)-ATP(RB1)
4	5	1	0.23900	75000 ; ATP(RB1)-ATP(RB2)
5	6	1	0.34300	75000 ; ATP(RB2)-ATP(SC1)

[constraints]

6	7	1	0.239	; Adenosine SC1-SC2
6	9	1	0.163	; Adenosine SC1-SC4
7	8	1	0.272	; Adenosine SC2-SC3
8	9	1	0.305	; Adenosine SC3-SC4

[exclusions]

1 3 ; may be removed in case of for Martini2p

[angles]

; Backbone angles

1	2	3	10	133	30	; ATP(PG)-ATP(PB)-ATP(PA)
2	3	4	10	114.5	25	; ATP(PB)-ATP(PA)-ATP(RB1)
3	4	5	10	80	10	; ATP(PA)-ATP(RB1)-ATP(RB2)
4	5	6	2	75	300	; ATP(RB1)-ATP(RB2)-ATP(SC1)
5	6	7	2	120	180	; ATP(RB2)-ATP(SC1)-ATP(SC2)
5	6	9	2	127.6	185	; ATP(RB2)-ATP(SC1)-ATP(SC4)
6	7	8	2	84.5	100	; Adenosine SC1-SC2-SC3
6	9	8	2	89.5	100	; Adenosine SC1-SC4-SC3
7	6	9	2	114	100	; Adenosine SC2-SC1-SC4
7	8	9	2	72	100	; Adenosine SC2-SC3-SC4

[dihedrals]

1	2	3	4	1	0	2.5	1 ; ATP(PG)-ATP(PB)-ATP(PA)-ATP(RB1)
3	4	5	6	2	-100	50	; ATP(PA)-ATP(RB1)-ATP(RB2)-ATP(SC1)
4	5	6	7	1	-95	4	2 ; ATP(RB1)-ATP(RB2)-ATP(SC1)-ATP(SC2)
4	5	6	9	1	-30	5	2 ; ATP(RB1)-ATP(RB2)-ATP(SC1)-ATP(SC4)
4	5	6	7	2	-180	2	; ATP(RB1)-ATP(RB2)-ATP(SC1)-ATP(SC2)
4	5	6	9	2	80	0.5	; ATP(RB1)-ATP(RB2)-ATP(SC1)-ATP(SC4)
6	7	8	9	2	0	40	; Adenosine SC1-SC2-SC3-SC4

#ifdef POSRES

#ifndef POSRES_FC

```
#define POSRES_FC 2000.00
#endif
[ position_restraints ]
  1   1   POSRES_FC   POSRES_FC   POSRES_FC
  2   1   POSRES_FC   POSRES_FC   POSRES_FC
  3   1   POSRES_FC   POSRES_FC   POSRES_FC
  4   1   POSRES_FC   POSRES_FC   POSRES_FC
  5   1   POSRES_FC   POSRES_FC   POSRES_FC
#endif
```

2. page 6, (kcal/mol)/Å² should be kcal mol⁻¹ Å⁻²

Response: Done in the revised manuscript.

3. In the main text, when a figure is referred, figure X should be Figure X.

Response: Done in the revised manuscript.

4. In Figure S2, the unit of RMSD should be angstrom, but not “A”, though we can use “A” for angstrom in informal documents.

Response: Done in Figure S2.

Again, we would like to thank Reviewer #2 for identifying the errors and providing constructive advice, which have helped us improve our paper.